# Design Implementation and Evaluation of a Mobile Continuous Blood Oxygen Saturation Monitoring System

**DOI:** 10.3390/s20226581

**Published:** 2020-11-18

**Authors:** Qingxue Zhang, David Arney, Julian M. Goldman, Eric M. Isselbacher, Antonis A. Armoundas

**Affiliations:** 1Cardiovascular Research Center, Massachusetts General Hospital, Boston, MA 02129, USA; qxzhang@iu.edu; 2The Electrical & Computer Engineering Department, Indiana University-Purdue University at Indianapolis, Indianapolis, IN 46202, USA; 3MD PnP Interoperability and Cybersecurity Program, Massachusetts General Hospital, Boston, MA 02114, USA; darney@mgh.harvard.edu (D.A.); JMGOLDMAN@mgh.harvard.edu (J.M.G.); 4Healthcare Transformation Lab, Massachusetts General Hospital, Boston, MA 02114, USA; eisselbacher@mgh.harvard.edu; 5Institute for Medical Engineering and Science, Massachusetts Institute of Technology Cambridge, Cambridge, MA 02139, USA

**Keywords:** blood oxygen saturation, wearable computer, mobile health, physiological signal processing

## Abstract

Objective: In this study, we built a mobile continuous Blood Oxygen Saturation (SpO2) monitor, and for the first time, explored key design principles towards daily applications. Methods: We firstly built a customized wearable computer that can sense two-channel photoplethysmogram (PPG) signals, and transmit the signals wirelessly to smartphone. Afterwards, we explored many SpO2 model building principles, focusing on linear/nonlinear models, different PPG parameter calculation methods, and different finger types. Moreover, we further compared PPG sensor placement principles by comparing different hand configurations and different finger configurations. Finally, a dataset collected from eleven human subjects was used to evaluate the mobile health monitor and explore all of the above design principles. Results: The experimental results show that the root mean square error of the SpO2 estimation is only 1.8, indicating the effectiveness of the system. Conclusion: These results indicate the effectiveness of the customized mobile SpO2 monitor and the selected design principles. Significance: This research is expected to facilitate the continuous SpO2 monitoring of patients with clinical indications.

## 1. Introduction

Photoplethysmography (PPG) is a convenient, low-cost technology that has been applied to various aspects of cardiovascular monitoring, such as blood oxygen saturation, heart rate, respiration, blood pressure, cardiac output, microvascular blood flow, endothelial function, arterial aging, and autonomic function [1]. The optical absorption of hemoglobin is a function of oxygenation and optical wavelength, so the use of PPG at multiple wavelengths is routinely used for measuring peripheral oxygen saturation (SpO2). SpO2 monitoring is an important vital sign in evaluating and monitoring the cardio-pulmonary function of both in-hospital and out-of-hospital patients [2,3,4]. As the average age of the US population increases and chronic conditions are becoming more prevalent, there is a need to improve the effectiveness of disease prevention, to enhance access to healthcare, and to sustain healthy independent living [5,6,7,8]. Mobile health technologies are expected to function not only as monitoring devices, but also as essential components in the delivery of healthcare to those with chronic diseases [6,7,8,9,10,11,12,13,14,15,16,17,18], especially in under-served populations. However, currently, there are no methods that are both effective and convenient to continuously monitor SpO2 in ambulatory patients. 

Prior studies on PPG-based SpO2 monitoring usually relied on inconvenient methods, such as wired solutions, or wireless methods [19,20,21,22], that did not explore key design principles, such as different finger types, fingers, and hands [23,24,25]. Furthermore, they did not investigate how different SpO2 models affect the estimation performance of SpO2. 

In this study, we have developed a highly convenient, continuous SpO2 monitor, and explored all the above mentioned key design principles. To the best of our knowledge, it is the first time a study thoroughly investigates all these key aspects towards medical-grade, continuous, wireless SpO2 monitoring. More specifically, our contributions include: (i) The building of a two-channel PPG wearable prototype for wireless optical signal sensing, (ii) the investigation of three finger types for PPG model calibration: Index, middle, and ring fingers, (iv) the investigation of different PPG parameter calculation methods for the PPG model building: Filtering-based approach and FFT-based approach, (v) the probe of different SpO2 estimation models: From linear and non-linear regressions models to the machine learning model, (vi) the comparison of left and right hands, and the comparison of different fingers, for SpO2 monitoring.

## 2. Methods

### 2.1. Human Study

The human studies were ethically approved by the institutional review board of the Massachusetts General Hospital (Protocol#: 2018P003132), and all participants provided oral informed consent. All studies were performed in accordance with relevant guidelines and regulations.

To evaluate the wearable computer prototype and the signal processing algorithms, we have collected data from eleven (M/F: 10/1) healthy human subjects of: 39 ± 17 years old, height 5.7 ± 0.3 ft., and weight 169 ± 22 lbs. During data collection, the PPG finger-cuff was placed on the index/middle/ring fingers on the left and right hand, respectively, while subjects were still in a sitting position. Thus, for each participant, there were six recordings, each one being 1 min long. 

### 2.2. Data Recording Equipment

We have developed a state-of-the-art signal acquisition, display, and processing system, which supports the acquisition, display, and real-time analysis of 16 analog signals sampled at 1000 Hz by a multi-channel 16-bit data acquisition card (National Instruments M-Series PCI6255) [26,27,28]. This system consists of custom software written in LabView 8.5 (National Instruments, Austin, TX, USA) and MATLAB 7.6 (MathWorks, Natick, MA, USA).

Analog PPG signals were acquired from a Tram-rac 4A (Marquette) and a SOLAR 8000 (Marquette) patient monitoring system as well as the Masimo SpO2 finger cuff and used as each subject’s gold-standard SpO2 measurement. For each finger, we firstly performed a gold-standard SpO2 measurement, and then, immediately thereafter, performed the wearable, battery-powered device (finger cuff, Texas Instruments; Model: AFE4490) measurement to minimize the chance of SpO2 fluctuations. 

### 2.3. Wearable Prototyping for Wireless Optical Sensing

In this section, the mobile SpO2 monitor prototyping and design principles are detailed. We have built a wireless prototype for two-channel PPG sensing, as shown in Figure 1. The sensor has an analog front-end (AFE) AFE4490 that is equipped with a Digital-to-Analog Converter (DAC) to drive the light-emitting device (LED) to emit light to the skin, as well as a high precision Analog-to-Digital Converter (ADC) to capture two-channel PPG signals. Then the sensed signals are parsed by a microcontroller unit (MCU) MSP430, and sent to a TivaC Launchpad which triggers a Bluetooth (BT) module HC-05 for wireless data transmission. The MCU communicates with the AFE via the serial peripheral interface (SPI) and with the BT via the universal asynchronous receiver-transmitter (UART). The software flow of the MCU is shown in Figure 2A, where the MCU configures the ADC, DAC, and BT at the beginning, and then listens to the ADC to parse the data and send the data to BT.

### 2.4. Mobile App Development

Thereafter, we built an app to calculate the SpO2 in real-time, as shown in Figure 2B,C. This app is built on an Android OS, and developed using the Java language. It configures the Bluetooth module on the smartphone to listen to transmission events from the wearable computer, parses the PPG data, estimates the SpO2 values, and visualizes both the PPG and SpO2 signals (Figure 2B). 

More specifically, the app receives two channel PPG signals (red and infrared) continuously sent by the wearable monitor. The app then performs the SpO2 calculation based on the algorithms described below. The estimated SpO2 value is displayed on the phone screen in real-time (Figure 2C). Both raw PPG signals and the calculated SpO2 values are simultaneously stored to the Secure Digital (SD) card on the phone. Therefore, the developed phone app can receive, process, store, and visualize the data.

### 2.5. System Calibration

To calibrate the system, i.e., to train the PPG model parameters, we have collected PPG signals using a Fluke (Fluke Biomedical, ProSim8) SpO2 tester, which generates artificial PPG signals in a plastic finger according to a predetermined configuration. We have considered three figure type configurations: Index, middle, and ring fingers. These three finger types correspond to different pre-stored R-curves in the Fluke tester, aiming to provide corresponding PPG signals for calibration purposes. We have collected and used PPG signals corresponding to different SpO2 values in order to determine the SpO2 model parameters. We have configured the Fluke to deliver SpO2 values ranging from 70% to 100% and then used our wearable device to collect the simulated PPG signals. The emulated PPG signals and the SpO2 values were used to calibrate different SpO2 models (introduced below). Combing the finger types with the four SpO2 models to be introduced later, there are in total 12 SpO2 models to be calibrated and compared. 

Pulse Oximeter testers like the Fluke ProSim-8 measure the output of the pulse oximeter’s LEDs and recreate the light modulated as though it had passed through a human finger. These testers store R-curves calibrated for particular brands of pulse oximeters and use these curves to create light output that will cause a pulse oximeter of the matching brand to output an SpO2 value corresponding to the tester’s settings. They are used by biomedical engineers to test individual pulse oximeters in clinical settings to confirm that each device under test shows the same SpO2 output as the reference device that was used to create the R-curve. They cannot be used to check the accuracy or calibration of a pulse oximeter, instead, they are ‘transfer standards’ that compare the device under test to a reference device that was calibrated by other means, usually a desaturation study with human volunteers. Using a pulse oximeter tester to calibrate a novel pulse oximeter allows the developers to match their R-curve to the manufacturer’s chosen in the tester, but because they use different LEDs and photosensors and likely different filtering algorithms, this does not ensure accuracy for human use.

### 2.6. PPG Parameter Estimation

SpO2 is defined as,
(1)$$SpO_{2}=\frac{\rho^{HbO_{2}}}{\rho^{HbO_{2}}+\rho^{Hb}}$$
where, $\rho^{HbO_{2}}$ and $\rho^{Hb}$ are the concentration of oxyhemoglobin and hemoglobin, respectively. 

To estimate SpO2 from two-channel PPG signals, we firstly calculate a key parameter, the ratio of ratios R as,
(2)R=ACRedDCRedACIRDCIR
where, $AC_{Red}$ and $DC_{Red}$ are the AC and DC components of the red PPG signal, respectively, and $AC_{IR}$ and $DC_{IR}$ are the corresponding components of the infra-red (IR) PPG signal. 

We have compared two approaches to estimate the parameter R: A filtering method and an FFT method [29]. For the filtering method, a high-pass filter with a cut-off frequency of 0.2 Hz is applied to the red and infra-red PPG signals, respectively, to remove the direct current (DC) component, yielding two quasi-sinusoid waveforms that correspond to the AC components of the red PPG channel and the infra-red PPG channel, respectively. The DC component for each PPG channel is calculated by subtracting its AC component from the original PPG signal without filtering. The peak-to-peak voltage of these two waveforms are the alternating current (AC) components, $AC_{Red}$ and $AC_{IR}$. The average voltage level values of each waveform before the low-pass filter are the DC components, $DC_{Red}$ and $DC_{IR}$. For the FFT method, the DC and AC components of the PPG signal are transformed from the time domain to the frequency domain. More specifically, the DC component of the PPG signal is calculated as the zero-frequency component of the FFT spectrum. The AC component is determined from the remaining FFT spectrum as the component with the maximum magnitude. Following derivation of $AC_{Red}$, $AC_{IR}$, $DC_{Red}$ and $DC_{IR}$, the parameter R is derived using Equation (2).

Since oxyhemoglobin (HbO2) has a much higher attenuation coefficient at the red-light wavelength compared to that of the IR light, and hemoglobin (Hb) has a higher attenuation coefficient for the IR light, the key parameter R reflects the concentration of HbO2 and Hb. It is reported that R is significantly correlated with SpO2 and thus can be used to estimate SpO2 noninvasively. 

### 2.7. SpO2 Estimation Models

Several models have been considered in the SpO2 estimation, such as linear/quadratic/cubic polynomials, and a regression tree: (3)$$SpO_{2}=a*R+b$$
(4)$$SpO_{2}=a*R^{2}+b*R+c$$
(5)$$SpO_{2}=a*R^{3}+b*R^{2}+c*R+d$$
(6)$$SpO_{2}=RegressionTree\left(R\right)$$

We aim to evaluate how the complexity of the SpO2 model affects the SpO2 estimation accuracy. While the linear polynomial has been used in prior studies [20,25], a nonlinear model is likely to better capture the complex relation between the parameter R and SpO2. Therefore, here, we thoroughly study and compare four different SpO2 models to determine the most appropriate one.

### 2.8. Finger Types in Model Calibration

We further take into consideration different finger types (index, middle, and ring), in building the SpO2 model. In the calibration phase, we aim to select the best combination of SpO2 model, finger type and approach for PPG parameter estimation. The selected combination will be finally used to evaluate the SpO2 estimation performance in the human study.

### 2.9. Inter-Hand and Inter-Finger Model Evaluation in Humans

In a study in humans, we first evaluated the SpO2 performance by comparing the left hand and right hand, and reported the root mean square error (RMSE, %), respectively, using the Wilcoxon signed-rank test.

We have further evaluated the calibrated model performance across fingers (index/middle/ring), using the Wilcoxon signed-rank test. 

## 3. Results

### 3.1. Calibration

In Figure 3, we present the comparison of the three finger types using the Fluke simulator, employing the four types of models we have proposed (above), totaling 12 different kinds of combinations. Md1/2/3/4, in Figure 3 represent: Model based first-order polynomial, second-order polynomial, third-order polynomial, and decision tree, respectively. 

With respect to the finger type, we observe that the middle finger type is superior to the ring and index finger types, exhibiting the lowest RMSE of 2.2%–2.8%. The box plots also show that the middle finger type has a much lower error than the other two finger types. The ring finger and the index finger have an RMSE as high as 43.1% and 46.5%, respectively. Therefore, the middle finger type is selected to calibrate the SpO2 model. 

With respect to the model type, both model 2 and model 3 correspond to the lowest RMSE, but considering model 2 is of lower complexity and therefore it is easier to implement in the smartphone, we finally choose model 2 as the final SpO2 model. Outliers are likely to be due to high sensitivity to finger movement, and therefore poor contact and signal quality. This is very common for PPG signal acquisition since even a small finger movement is expected to cause sensor to skin contact changes. 

We have further compared two methods to estimate the PPG parameter R, i.e., the ratio of ratios. As shown in Figure 4, the RMSE of the filtering method is 2.2%, much lower than that of the FFT method, 6.1. Therefore, the filtering method is chosen to be the parameter calculation method. It is worth noting that the filtering method is used to calculate the parameter ***R***, which is fed into the SpO2 model for SpO2 estimation. Therefore, this calibration step involves the method to calculate ***R***, not to train parameters for the SpO2 model training. Therefore, in the calibration step, we have selected the optimal model as: Middle finger type, model 2 (second-order polynomial function), and filtering method-based parameter ***R*** calculation. 

We have found that the mean and standard deviation of the R value for the middle finger, to be 0.58 and 0.04, respectively.

### 3.2. Inter-Hand Evaluation

We used the calibrated SpO2 model, i.e., the middle finger type, second-order polynomial, and the filtering-based parameter calculation method, to measure the SpO2 on 11 human subjects. 

Firstly, we compared the SpO2 model between hands. For each hand, we tested the model on three fingers: Index, middle, and ring fingers, and then averaged the performance across fingers. The RMSE (%) of the right and left hands (1.9 and 2.5, respectively), shown in Figure 5, indicates that there are no significant differences between the two hands (*p* = 1.00). This observation indicates that, as expected, both hands have similar blood perfusion conditions and are almost equivalent in terms of SpO2 measurement, which demonstrates the effectiveness of the calibrated SpO2 model.

### 3.3. Inter-Finger Evaluation

To further evaluate the SpO2 model, we have compared its performance for different fingers, as shown in Figure 6. The RMSE (%) of three fingers ranges from 1.5 to 3.4, indicating the error is very small (*p* = 0.13). This has demonstrated that the calibrated SpO2 model can be used on not only different hands but also different fingers. Moreover, the RMSE (%) of the left index finger is as low as 1.8, much lower than the FDA standard that requires the RMSE (%) to be less than 3. Overall, we observe that both index and middle fingers can be used in mobile SpO2 monitoring. 

We recognize that anatomical differences among fingers may result in different contact conditions within the probe, which may induce artifact and outliers, as seen in Figure 3. However, the overall performance of this system indicates a potentially reliable monitoring device. 

## 4. Discussion

Despite the availability of a multitude of evidence-based therapies for the treatment of chronic diseases, the burden of these diseases on the US population remains unacceptably high, with an estimated one million admissions per year, primarily involving the elderly [30]. At the same time, there is increased availability of new technologies and an ever-improving health information technology infrastructure with >90% of American adults owning a cell phone and 55% having a smartphone [9,31]. The evolution of these wireless devices is expected to mark a new era in medicine and a transition from population-level health care to individualized medicine. 

In this paper, we introduce a wireless, smartphone-based, continuous SpO2 monitoring system that aims to supplement cvrPhone, a medical-grade, smartphone-based cardiac and respiratory monitoring platform [6,7,8]. To avoid an invasive measurement directly from a blood sample, we have researched and developed a PPG-based estimation approach. Using the transmitted optical light from one side of the finger (quasi-periodic PPG signals) resulting from a beam of light emitted by an LED on the other side of the finger [23], one can monitor blood flow fluctuations in microvascular beds at peripheral body sites (i.e., in the finger). Then, using two different wavelengths, the SpO2 can be estimated from the two-channel PPG readings.

There are several major findings in this study. First, our two-channel PPG wearable device can robustly acquire PPG signals for SpO2 estimation, second, in terms of SpO2 model calibration, the middle finger (the most common finger type) is more appropriate than the index or ring finger types, third, the filtering-based method is very effective to calculate the ratio of ratios—A SpO2-relevant parameter, fourth, the second-order polynomial model provides the best performance, fifth, our wearable device performs equally well for both hands and different fingers in measuring the SpO2. 

Current SpO2 technologies [23,24,25] usually utilize wired methods [32] that are inconvenient for mobile daily use. Studies that have employed wireless methods [19,20,21,22] have not explored critical design principles, including different model calibration methods, nonlinear versus linear models, and placement of device on different hands and fingers. Therefore, this study has filled this knowledge gap and provided valuable findings on mobile SpO2 monitor design.

In conclusion, SpO2 measurement is an important vital sign of cardiopulmonary monitoring, not only in a routine in-hospital assessment of patients’ physiology, but also in out-of-hospital follow-up patients’ treatment. Ambulatory, Bluetooth/smartphone-based monitoring of SpO2, together with 12 lead ECG, respiration rate, and tidal volume, provide to the end-user (both the patient and patient-care-team) the benefits of: (i) Extensive, continuous respiratory monitoring never before possible to the non-ventilated patient to assess the incidence, severity, and progression of chronic conditions, (ii) assisting patients and clinicians in the immediate evaluation of the patient’s respiratory system and changes in respiratory state that can precede respiratory depression and death, (iii) the ability to quickly assess an underlying ischemic episode and an-arrhythmic risk in a trend graph and assist patients and clinicians in determining whether immediate therapy is needed in order to prevent life-threatening arrhythmias and death, (iv) it will permit the care-team to constantly monitor the high volume of patients under their care. 

**Study Limitations** An important aspect of the present study is that the evaluation of the proposed mobile, continuous blood oxygen saturation monitoring system, presently, has been evaluated only in normal subjects, with a limited SpO_2_ range (95%–100%). Another limitation involves the use of a commercial device as the gold standard and not a blood test. Outliers due to high sensitivity to finger movement may affect contact and signal quality, therefore, an improved probe design is expected to reduce sensor to skin contact changes. Finally, the small number of tested subjects limits the power of this study.

## Figures and Tables

**Figure 1 sensors-20-06581-f001:**
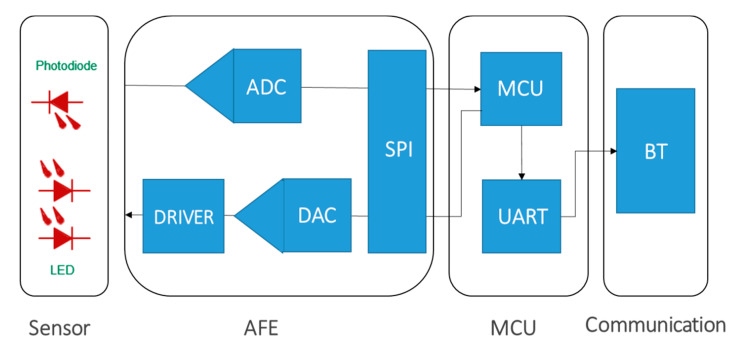
The wearable sensor for 2-channel photoplethysmogram (PPG) sensing. It has an analog front end (AFE) that includes a digital-to-analog converter (DAC) to drive light-emitting devices (LEDs) to emit light to the skin, and an analog-to-digital converter (ADC) to sense the light after skin absorption (transmission mode). It also includes a microcontroller unit (MCU) that communicates with the AFE via a serial peripheral interface (SPI) and with the Bluetooth (BT) module via a universal asynchronous receiver-transmitter (UART).

**Figure 2 sensors-20-06581-f002:**
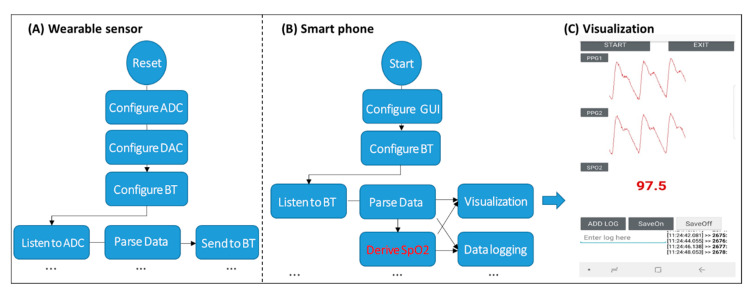
The mobile oxygen saturation (SpO_2_) system that includes a wearable sensor for 2-channel photoplethysmogram (PPG) signal sensing and a smartphone application (APP) for data analysis and visualization. The sensor leverages a digital-to-analog converter (DAC) and an analog-to-digital converter (ADC) to emit light to and sense residual light (PPG signals) from the skin. The smartphone app configures the graphical-user-interface (GUI) and Bluetooth (BT) module to receive the data from the sensor and analyze the data. (**A**) Wearable sensor, (**B**) Smart phone, (**C**) Visualization.

**Figure 3 sensors-20-06581-f003:**
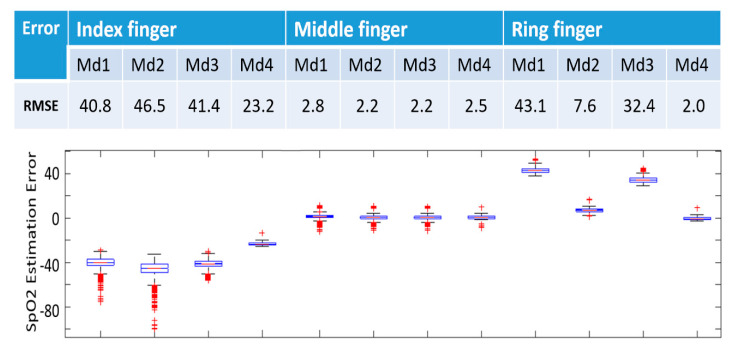
SpO2 model calibration of different finger types and SpO2 model types, showing that the middle finger type and the Md2 model are the best combination to calibrate the SpO2 model. RMSE: Root mean square error (%), Md1/2/3/4: Model based on first-order polynomial, second-order polynomial, third-order polynomial, and decision tree, respectively. Bar graphs represent 10, 25, 50, 75, and 90 percentiles of SpO2 estimation error. Md: Model.

**Figure 4 sensors-20-06581-f004:**
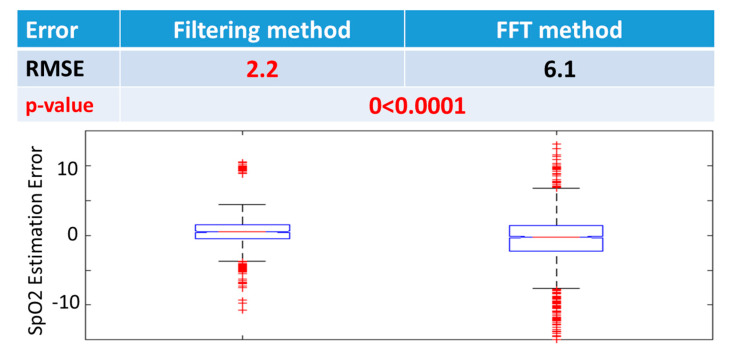
SpO2 model calibration on different parameter calculation methods (filtering method and FFT-based method), showing that the filtering method provides the best performance (*p* < 0.0001, sign-rank test). RMSE: Root mean square error (%), FFT: Fast Fourier Transform. Bar graphs represent 10, 25, 50, 75, and 90 percentiles of SpO2 estimation error.

**Figure 5 sensors-20-06581-f005:**
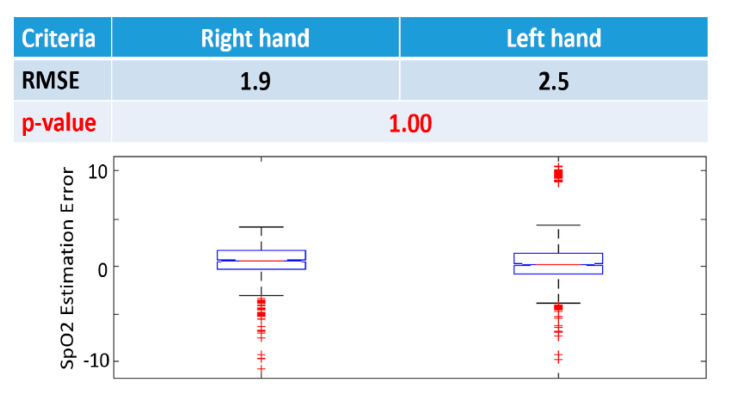
SpO2 model validation on different hands, showing that there is no significant difference between two hands. RMSE: Root mean square error (%), *p*-value is calculated using Wilcoxon signed-rank test. Bar graphs represent 10, 25, 50, 75, and 90 percentiles of SpO2 estimation error.

**Figure 6 sensors-20-06581-f006:**
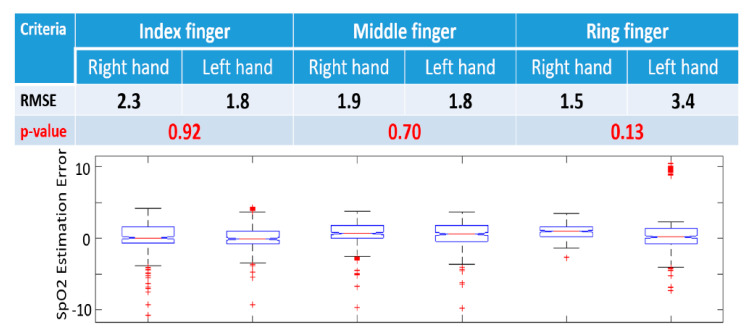
SpO2 model validation on different fingers (index, middle, and ring fingers), indicating that there is no significant difference among these fingers, and that the index and middle fingers are more appropriate for SpO2 monitoring because of lower RMSE. RMSE: Root mean square error (%), *p*-value is calculated using Wilcoxon signed-rank test. Bar graphs represent 10, 25, 50, 75, and 90 percentiles of SpO2 estimation error.

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
