# Peer review of "Design Implementation and Evaluation of a Mobile Continuous Blood Oxygen Saturation Monitoring System"

_sensors, 2020, doi:10.3390/s20226581_

Round 1
Reviewer 1 Report
A well written article overall. There were a few typos which easily be sorted.
The filter method mentions a low low pass filter stage is used to remove the DC component although I think this should be a high pass filter stage? To extract / separate the AC and DC then a mix of high and low pass filters could be used?
I am not clear on the many outliers on the results plots for some fingers? What are these ?noisy beats and if so when comparing hand sites are your results being highly skewed by artefact rather than true pulses. I really don't understand the big differences between fingers? For example, Figure 3 index finger - lots of outliers here for only 2 of the 4 models? Why is this if the outliers are noise related? The right and left sides are similar which I would expect but the outliers appear to disappear?
Suggest on Figure 3 - put this in order of Index Finger through to Ring finger.
Make it clear what md1-4 relate to in terms of named equations in the text.
Figure 3 seems a puzzle it says md2 better than md3? But noting Ring Finger md4 has the lowest value overall.
I think I have misunderstood something fundamental here - and if I di then the wider readership will also likely do so. The outliers are puzzling and sometimes speckled to one side of the median? Maybe a new figure with example traces for a good clean pulse and low RMSE against one that has a very high RMSE value would help, along with text, to understand what is happening. Sugest check all calculations and summary stats that went to produce Figure 3, as a minimum check that is.
What were the typical R values and coefficients used in the models - even the best fit model used as an example.
Define the filter cut-off frequencies used?
Define the FFT methods used?
Define any units of RMSE on all results.
Suggest include a citation to the Webster edited book on Pulse Oximetry - there is some good discussion and information on pulse oximeter calibration as general background which would compliment this article.
Reviewer 2 Report
The article describes the design and initial testing of a mobile continuous pulse oximeter. The design was done using a plastic finger SPO2 tester and was checked on several finger types to find the one with lowest error.
The article nicely describe the process of design and method choosing, and nicely describes the potential advantages of the proposed device.
However there are some important limitations in the article.
Major comments:
1. Like mentioned by the authors, an important limitation of the study is the fact that the device was tested only on people with saturation of 95-100%. I am concerned that this underlines the assessment of mean error. For example- a device that will constantly display the number 97% without checking saturation at all will have a low mean error of less than 3%.
I think that because of that major limitation the word "validation" should be deleted from the article title. and a comment should be made in the abstract that the FDA requirement of mean error of less than 3% is in the range of 70-100% and the test was done only for the range of 95-100%.
2. Another important limitation is that each test was done only for a minute, and without mentioning if the subjects moved there hands or sat still. An important aspect of a wearable device is it's ability to provide accurate reading for prolonged periods, during movement, and in different positions.
Minor comments:
1. It will be helpful to provide a picture of the actual device, if possible.
2. Please delete in line 44 also reference, all 3 Android based papers
Round 2
Reviewer 1 Report
The revised version is improved. I still have some minor comments which need to be incorporated.
- Line 199 - reword as could be read 2 different ways - it is about the little finger or is it since there is little finger movement (and so on) ?
- RMSE for SO2 to be in % in all results / text / tables / figures.
- Should there be p value on Figure 4 (comparing groups), as per other figures and comparisons?
- I believe there will have been noisy recordings made ?for index finger site and hence the outliers prominent here. The authors response document gives a nice figure (3) comparing a clean trace with a noisy trace. Artefact - movement etc can be a major problem with PPG. Now I understand the methods more from the revision - the topic of noise needs to be highlighted at least in the Discussion regarding the outliers. This comment could lead into probe design - and I mean the attachment method which is a fundamental consideration in a PPG system. This needs to be said in the Discussion - it will be helpful for the reader.
